# *n*-Butene Synthesis in the Dimethyl Ether-to-Olefin Reaction over Zeolites

**Toshiaki Hanaoka [1,*], Masaru Aoyagi [1] and Yusuke Edashige [2]**

1 Organic Materials Diagnosis Group, Research Institute for Sustainable Chemistry, National Institute of Advanced Industrial Science and Technology (AIST), Hiroshima 739-0046, Japan; masaru-aoyagi@aist.go.jp

2 Faculty of Agriculture, Ehime University, Matsuyama 790-8566, Japan; yedash@agr.ehime-u.ac.jp

* Correspondence: t.hanaoka@aist.go.jp

**Abstract:** Zeolite catalysts that could allow the efficient synthesis of *n*-butene, such as 1-butene, *trans*-2-butene, and *cis*-2-butene, in the dimethyl ether (DME)-to-olefin (DTO) reaction were investigated using a fixed-bed flow reactor. The zeolites were characterized by $N_2$ adsorption and desorption, X-ray diffraction (XRD), thermogravimetry (TG), and $NH_3$ temperature-programmed desorption ($NH_3$-TPD). A screening of ten available zeolites indicated that the ferrierite zeolite with $NH_4^+$ as the cation showed the highest *n*-butene yield. The effect of the temperature of calcination as a pretreatment method on the catalytic performance was studied using three zeolites with suitable topologies. The calcination temperature significantly affected DME conversion and *n*-butene yield. The ferrierite zeolite showed the highest *n*-butene yield at a calcination temperature of 773 K. Multiple regression analysis was performed to determine the correlation between the six values obtained using $N_2$ adsorption/desorption and $NH_3$-TPD analyses, and the *n*-butene yield. The contribution rate of the strong acid site alone as an explanatory variable was 69.9%; however, the addition of micropore volume was statistically appropriate, leading to an increase in the contribution rate to 76.1%. Insights into the mechanism of *n*-butene synthesis in the DTO reaction were obtained using these parameters.

**Keywords:** dimethyl ether; *n*-butene; zeolite; strong acid; micropore

## 1. Introduction

1,3-Butadiene (1,3-BD) is a raw material for general-purpose rubbers such as polybutadiene and styrene-butadiene rubbers. The demand is expected to increase in the future because its addition to rubber and plastic can improve durability and flexibility. In 2018, approximately 15.3 billion kilograms of synthetic rubber was produced worldwide [1].

Currently, 1,3-BD is mainly produced as a co-product in the production of ethylene from naphtha. In recent years, the amount of ethylene produced from shale gas, which is cheaper than naphtha, has increased. As it is difficult to produce 1,3-BD from shale gas, there are concerns about the adequate supply. In recent years, 1,3-BD synthesis from biomass is desirable and has attracted considerable attention [2,3]. As biomass is the only renewable carbon resource, the industrialization of 1,3-BD production from biomass can significantly contribute to decarbonization.

Hanaoka et al. proposed a thermochemical process for producing 1,3-BD from biomass (Figure 1) and showed superior economics through simulation in comparison to its use in power generation [4,5]. The process comprised four unit operations: gasification, dimethyl ether (DME) synthesis, *n*-butene synthesis in the DME-to-olefin (DTO) reaction, and isomerization/dehydrogenation.

No catalyst has yet been reported that facilitates the efficient synthesis of *n*-butene in the DTO reaction. Hanaoka et al. reported a target *n*-butene yield (29.9 C-mol%) in which 1,3-BD synthesis showed economics that were comparable to those of power generation [6].

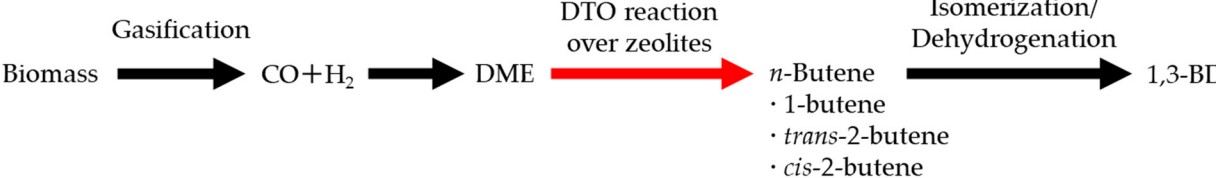

**Figure 1.** 1,3-Butadiene (1,3-BD) production process from biomass.

Zeolites have been reported as promising catalysts for DTO reactions [7]. Efficient propylene synthesis has been reported using zeolites with topologies such as AEI/CHA [8–10], CHA [11–13], MFI [14–18], MSE [19–22], EUO [23], and YFI [24]. Propylene is the main target product in the DTO reaction, with butene as a low-yield byproduct. The relationship between the physical properties of zeolites and *n*-butene yield remains unclear.

In this study, highly efficient zeolites were experimentally investigated for application in the DTO reaction. Promising zeolites were selected after screening. Using zeolites preheated at different calcination temperatures, the relationship between the catalytic performance and parameters such as crystal structure, surface area, and acidic properties were systematically studied. Then, strong acid site and micropore volume with high correlation to *n*-butene yield were identified statistically as having high correlation to *n*-butene yield. The experimental results can be explained using the selected parameters based on the DTO reaction and methanol-to-hydrocarbon (MTH) reaction mechanisms.

## 2. Results and Discussion

As a preliminary experiment, the DTO reaction was performed at 673 K using inert $SiO_2$ instead of zeolites. The DME conversion rate was 0.1%, while *n*-butene was not detected. This result indicates that the obtained *n*-butene can be attributed to the zeolite activity.

### 2.1. Screening of Zeolites

Ten zeolites were screened. Prior to the DTO reaction, the zeolites were calcined at 773 K for two hours in a muffle furnace. The zeolite species, topology, $SiO_2/Al_2O_3$ molar ratio, and catalytic performance in the DTO reaction are included in Table 1. The DME conversion and product yield depended significantly on the zeolite species. In particular, the 720NHA, as a ferrierite zeolite (denoted as 720NHA (FER)), showed the highest *n*-butene yield of 27.6 C-mol%. The obtained yields of butene, including that of *iso*-butene, reached 36.2 C-mol%, exceeding the highest previously reported butene yield (24.9 C-mol%) [19].

**Table 1.** Catalytic performance in DTO reaction using various zeolites.

| Material | Topology | $SiO_2/Al_2O_3$ (mol/mol) | DME Conv. (%) | Product Yield (C-mol%) | | | | | | | | |
|---|---|---|---|---|---|---|---|---|---|---|---|---|
| | | | | $C_1$–$C_3$ | $C_2^=$ | $C_3^=$ | MeOH | $C_4$ [1] | $i$-$C_4^=$ | $n$-$C_4^=$ [2] | $C_5$ | $C_{6\leq}$ |
| 720NHA | FER | 18.5 | 98.7 | 5.7 | 6.0 | 12.3 | 2.2 | 0.2 | 8.6 | 27.6 | 29.1 | 6.7 |
| 720KOA | | 18.5 | 0.6 | 0.2 | 0.0 | 0.0 | 0.3 | 0.0 | 0.0 | 0.0 | 0.0 | 0.0 |
| 820NHA | | 22.5 | 100.0 | 14.2 | 11.0 | 21.9 | 0.1 | 15.8 | 5.9 | 12.7 | 7.2 | 11.1 |
| 822HOA | | 23.9 | 100.0 | 14.6 | 8.8 | 17.9 | 0.0 | 17.6 | 5.8 | 12.1 | 8.1 | 15.0 |
| 840NHA | MFI | 39.0 | 100.0 | 13.6 | 7.4 | 13.7 | 0.0 | 14.8 | 7.1 | 13.4 | 9.2 | 20.7 |
| 840HOA | | 37.0 | 100.0 | 12.8 | 13.2 | 18.7 | 0.0 | 13.1 | 6.0 | 12.9 | 7.1 | 16.1 |
| 890HOA | | 2120 | 71.4 | 0.3 | 0.8 | 17.1 | 19.2 | 0.2 | 4.3 | 11.7 | 5.8 | 11.8 |
| 940HOA | BEA | 41.6 | 99.6 | 6.9 | 9.8 | 12.5 | 2.5 | 38.4 | 1.6 | 5.7 | 13.2 | 9.0 |
| 640HOA | MOR | 18.0 | 30.0 | 7.0 | 8.3 | 5.0 | 5.8 | 0.7 | 0.2 | 2.3 | 0.4 | 0.1 |
| 385HUA | FAU | 110 | 3.9 | 1.2 | 0.6 | 0.4 | 0.2 | 0.7 | 0.0 | 0.2 | 0.3 | 0.1 |

[1] *n*-butane + *iso*-butane, [2] 1-butene + *trans*-2-butene + *cis*-2-butene.

The parent 720NHA (FER) contained $NH_4^+$. The cation can be converted to $H^+$ through the desorption of $NH_3$ molecules upon calcination. The 720KOA (FER) with $K^+$ showed low DME conversion and *n*-butene yield, despite having the same $SiO_2/Al_2O_3$ molar ratio as 720NHA (FER) (Table 1). This result indicates that the cation type significantly affects the catalytic performance.

MFI zeolites constitute more available materials than other topologies. The four zeolites with the molar ratios of 22.5–39.0 showed 100% DME conversion and similar *n*-butene yields (12.1–13.4 C-mol%) (Table 1). In contrast, the 890HOA (MFI) with the ratio of 2120 showed a low DME conversion (71.4%) and high MeOH yield (19.2 C-mol%).

Al-Dughaither and de Lasa reported that in the DTO reaction employing the MFI zeolites, butene selectivity was independent of the $SiO_2/Al_2O_3$ molar ratio in the 30–280 range [25]. These results are consistent with the catalytic performance of the five MFI zeolites (Table 1). In the hydration of DME using MFI zeolites, no MeOH production was observed at 673 K due to thermodynamic disadvantages, although a high MeOH selectivity was reported at 473–573 K [26,27]. Magomedova et al. reported that in the DTO reaction at 593 K, MeOH was selectively produced at <70% DME conversion in the alkene cycle [28]. This finding is similar to the results in the present study. The high MeOH yield at high temperature (19.2 C-mol%) is an interesting finding, although it is not the focus of this study.

One material each of the BEA, MOR, and FAU zeolites was used, which showed low *n*-butene yields. 940HOA (BEA) resulted in almost 100% DME conversion and a very high yield of alkanes with four carbon atoms (38.4 C-mol%). The selective synthesis of saturated hydrocarbons was interesting for comparison with the results of 720NHA (FER).

## 2.2. Effects of Calcination Temperature

The *n*-butene yields were obtained in the order of 720NHA (FER) > 840NHA (MFI) > 940HOA (BEA) (Table 1). The calcination temperature was expected to affect not only the amounts of $NH_3$ and $H_2O$ molecules desorbed from the surface, but also the number of acid sites, acid strength, and pore structure. The relationship between the temperature and weight losses of 720NHA (FER), 840NHA (MFI), and 940HOA (BEA) was determined by thermogravimetry (TG) (Figure 2).

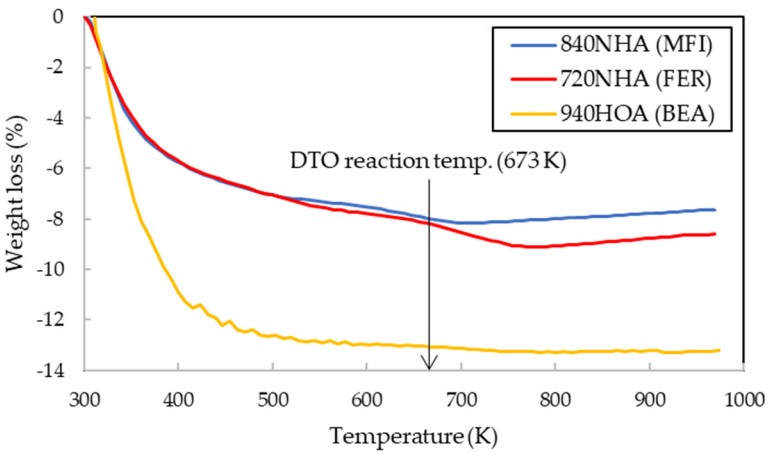

**Figure 2.** Dependence of weight loss on temperature by TG analysis.

For all three types of zeolites, with an increase in temperature to approximately 400 K, a drastic weight loss was observed due to the desorption of physically adsorbed $H_2O$ molecules. Thereafter, the weights of 840NHA (MFI) and 720NHA (FER) decreased slowly because of the desorption of $NH_3$ molecules, as well as dehydration, reaching their minimum values at approximately 700 K and 780 K, respectively, and then increased again with increasing temperatures. In contrast, the weight of 940HOA (BER) decreased monotonically, remaining almost constant above 760 K.

The catalytic performance was evaluated using 720NHA (FER), 840NHA (MFI), and 940HOA (BEA) (Table 2). The calcination temperatures were 723, 773, 823, and 873 K, and had significant effects on catalytic performance. Notably, 720NHA (FER) showed the highest *n*-butene yield (27.6 C-mol%) at 773 K, while a decrease in the *n*-butene yield was observed with increasing calcination temperatures for 840NHA (MFI) and 940HOA (BEA).

**Table 2.** Catalytic performance in DTO reaction using zeolites at different calcination temperatures.

| Material | Topology | Calcination Temp. (K) | DME Conv. (%) | Product Yield (C-mol%) | | | | | | | | |
|---|---|---|---|---|---|---|---|---|---|---|---|---|
| | | | | $C_1$–$C_3$ | $C_2^=$ | $C_3^=$ | MeOH | $C_4$ [1] | $i$-$C_4^=$ | $n$-$C_4^=$ [2] | $C_5$ | $C_{6\leq}$ |
| 720NHA | FER | N/A | 4.6 | 1.3 | 0.1 | 0.1 | 2.4 | 0.0 | 0.0 | 0.4 | 0.2 | 0.1 |
| | | 723 | 95.4 | 4.6 | 6.7 | 17.7 | 0.1 | 0.4 | 10.8 | 22.3 | 27.0 | 5.7 |
| | | 773 | 98.7 | 5.7 | 6.0 | 12.3 | 2.2 | 0.2 | 8.6 | 27.6 | 29.1 | 6.7 |
| | | 823 | 91.2 | 6.6 | 7.1 | 14.2 | 1.0 | 0.6 | 8.6 | 26.7 | 21.8 | 4.5 |
| | | 873 | 31.1 | 5.4 | 3.3 | 6.5 | 0.3 | 0.6 | 3.4 | 6.8 | 3.5 | 1.2 |
| 840NHA | MFI | N/A | 97.3 | 8.1 | 12.7 | 20.6 | 4.4 | 5.4 | 5.2 | 18.8 | 9.9 | 11.8 |
| | | 723 | 92.6 | 5.8 | 9.5 | 24.6 | 0.2 | 6.0 | 8.8 | 18.7 | 7.9 | 11.1 |
| | | 773 | 100.0 | 13.6 | 7.4 | 13.7 | 0.0 | 14.8 | 7.1 | 13.4 | 9.2 | 20.7 |
| | | 823 | 100.0 | 14.2 | 7.0 | 13.6 | 0.0 | 19.3 | 6.0 | 13.0 | 9.9 | 17.0 |
| | | 873 | 60.9 | 13.5 | 3.1 | 5.4 | 0.4 | 12.4 | 2.0 | 4.2 | 4.4 | 15.5 |
| 940HOA | BEA | N/A | 99.7 | 9.0 | 10.7 | 10.6 | 0.7 | 41.3 | 1.3 | 4.3 | 13.3 | 8.5 |
| | | 723 | 99.3 | 6.1 | 9.3 | 14.6 | 3.2 | 33.4 | 2.8 | 7.8 | 12.2 | 9.7 |
| | | 773 | 99.6 | 6.9 | 9.8 | 12.5 | 2.5 | 38.4 | 1.6 | 5.7 | 13.2 | 9.0 |
| | | 823 | 98.5 | 7.1 | 9.2 | 12.3 | 4.4 | 35.0 | 2.5 | 6.9 | 12.4 | 8.5 |
| | | 873 | 88.8 | 8.6 | 8.9 | 9.7 | 0.9 | 37.3 | 1.0 | 3.7 | 11.6 | 7.1 |

[1] *n*-butane + *iso*-butane, [2] 1-butene + *trans*-2-butene + *cis*-2-butene.

In the DTO reaction, product selectivity typically follows the Anderson–Schulz–Flory distribution [29]. The product yields for each carbon number are shown for the three zeolites at different calcination temperatures (Figure 3). Notably, all three zeolite types frequently showed the highest yields of hydrocarbons with four carbon atoms. The *n*-butene yield (green bar) is also shown in Figure 3. The *n*-butene yields constituted 70–80% of the hydrocarbons with four carbon atoms for 720NHA (FER) (Figure 3a), exceeding those for 840NHA (MFI) and 940HOA (BEA). Park et al. reported the selectivity order of pentene + hexene > butene > propylene for the methanol-to-olefin (MTO) reaction using commercial ferrierite zeolites [30]. ZSM-22 (TON) with medium-pore (10-ring) is excellent for the synthesis of hydrocarbons with four or more carbon atoms, but its highest yields are for hydrocarbons with five or six carbon atoms [31,32].

940HOA (BEA) showed a high yield of four-carbon hydrocarbons ($\geq$42 C-mol%) regardless of the calcination temperature (Figure 3c), whereas the *n*-butene yield was $\leq$6.9 C-mol%. The characterization of 940HOA (BEA) suggested properties that could result in low *n*-butene yields.

$N_2$ adsorption and desorption, X-ray diffraction (XRD), and $NH_3$ temperature-programmed desorption ($NH_3$-TPD) analyses were employed to characterize the zeolites used in the DTO reaction.

The $N_2$ adsorption–desorption isotherms for 720 NHA (FER) at 77 K are shown at different calcination temperatures in Figure 4. All $N_2$ adsorption–desorption isotherms were dependent on the zeolite species and corresponded to the IUPAC classification type I and IV composite types. Without calcination, no adsorption was observed at low relative pressure ($p/p_0$) values derived from the micropores. In contrast, adsorption was observed at a low $p/p_0$ with calcination, and the amount of $N_2$ adsorbed increased with an increase in $p/p_0$; it also increased with an increase in the calcination temperature. For 840NHA (MFI) and 940HOA (BEA), adsorption was observed at low $p/p_0$ for all zeolite species, and calcination resulted in an increase in the amount of $N_2$ adsorption (Figures S1 and S2).

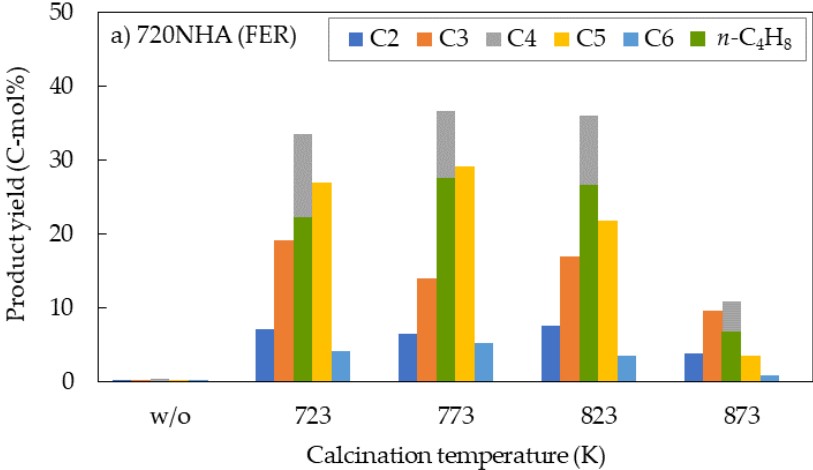

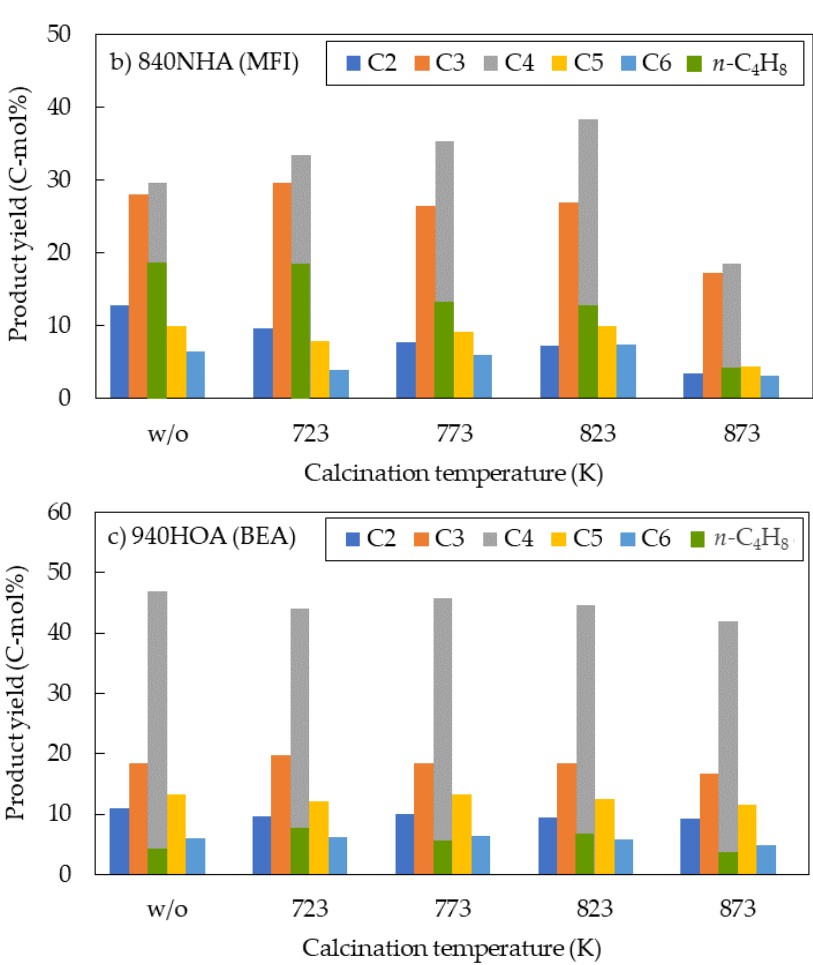

**Figure 3.** Relationship between calcination temperature and product yield for each carbon number: (**a**) 720NHA (FER); (**b**) 840NHA (MFI); and (**c**) 940HOA (BEA).

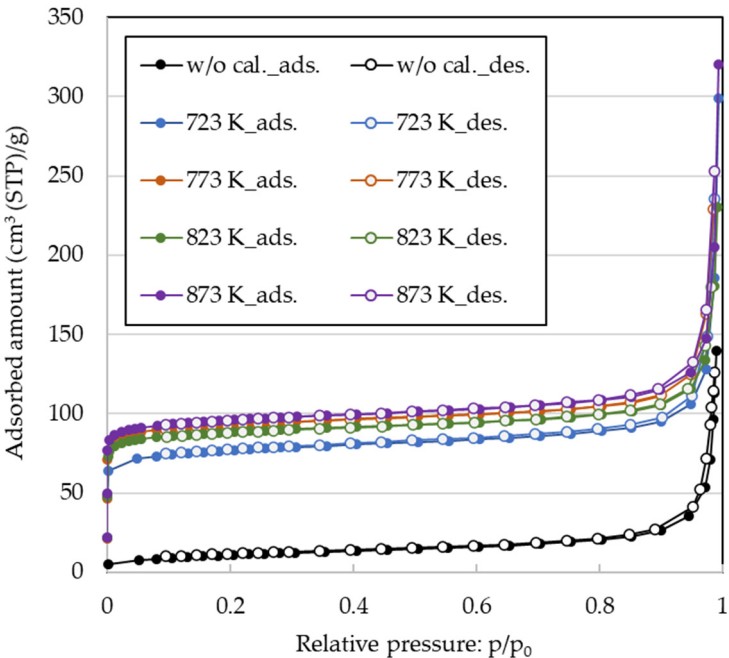

**Figure 4.** $N_2$ adsorption–desorption isotherms for 720NHA (FER) at 77 K.

The structural parameters of 720NHA (FER), 840NHA (MFI), and 940HOA (BEA) obtained from the $N_2$ adsorption–desorption isotherms are listed in Table 3. 940HOA (BEA) had the largest total surface area (757–823 m$^2$/g), micropore surface area (741–804 m$^2$/g), and micropore pore volume (0.24–0.26 cm$^3$/g). 720NHA (FER) showed the highest values for the external surface area (25–29 m$^2$/g) and total pore volume (0.21–0.38 cm$^3$/g). 840NHA (MFI) showed intermediate values among the three zeolites.

**Table 3.** Pore parameters for 720NHA (FER), 840NHA (MFI), and 940HOA (BEA) at different calcination temperatures.

| Material | Topology | Calcination Temp. (K) | Surface Area (m$^2$ g$^{-1}$) | | | Pore Volume (cm$^3$ g$^{-1}$) | |
|---|---|---|---|---|---|---|---|
| | | | Total | Micro [1] | External [1] | Total [2] | Micro [1] |
| 720NHA | FER | N/A | 35 | 10 | 25 | 0.21 | 0.01 |
| | | 723 | 381 | 353 | 28 | 0.37 | 0.11 |
| | | 773 | 487 | 460 | 27 | 0.37 | 0.14 |
| | | 823 | 462 | 435 | 27 | 0.33 | 0.13 |
| | | 873 | 503 | 474 | 29 | 0.38 | 0.14 |
| 840NHA | MFI | N/A | 471 | 469 | 2 | 0.15 | 0.14 |
| | | 723 | 522 | 519 | 3 | 0.17 | 0.17 |
| | | 773 | 529 | 525 | 4 | 0.18 | 0.17 |
| | | 823 | 461 | 458 | 3 | 0.16 | 0.16 |
| | | 873 | 503 | 495 | 8 | 0.17 | 0.16 |
| 940HOA | BEA | N/A | 757 | 741 | 16 | 0.30 | 0.24 |
| | | 723 | 773 | 758 | 15 | 0.30 | 0.25 |
| | | 773 | 806 | 789 | 16 | 0.33 | 0.26 |
| | | 823 | 786 | 768 | 18 | 0.32 | 0.25 |
| | | 873 | 823 | 804 | 19 | 0.34 | 0.26 |

[1] calculated using t-plot, [2] calculated using the Brunauer–Emmett–Teller method.

The XRD patterns for 720NHA (FER) at different calcination temperatures are shown in Figure 5. These results are in good agreement with the reported XRD pattern of the ferrierite zeolite [33,34], thereby confirming its structure. The diffraction peak at 9.6°, corresponding to the (200) reflection in a typical ferrierite structure, is slightly shifted

to approximately 9.3° (Figure 5b), supporting the presence of small micropores with a diameter of approximately 0.5 nm in the zeolite interlayer space [35].

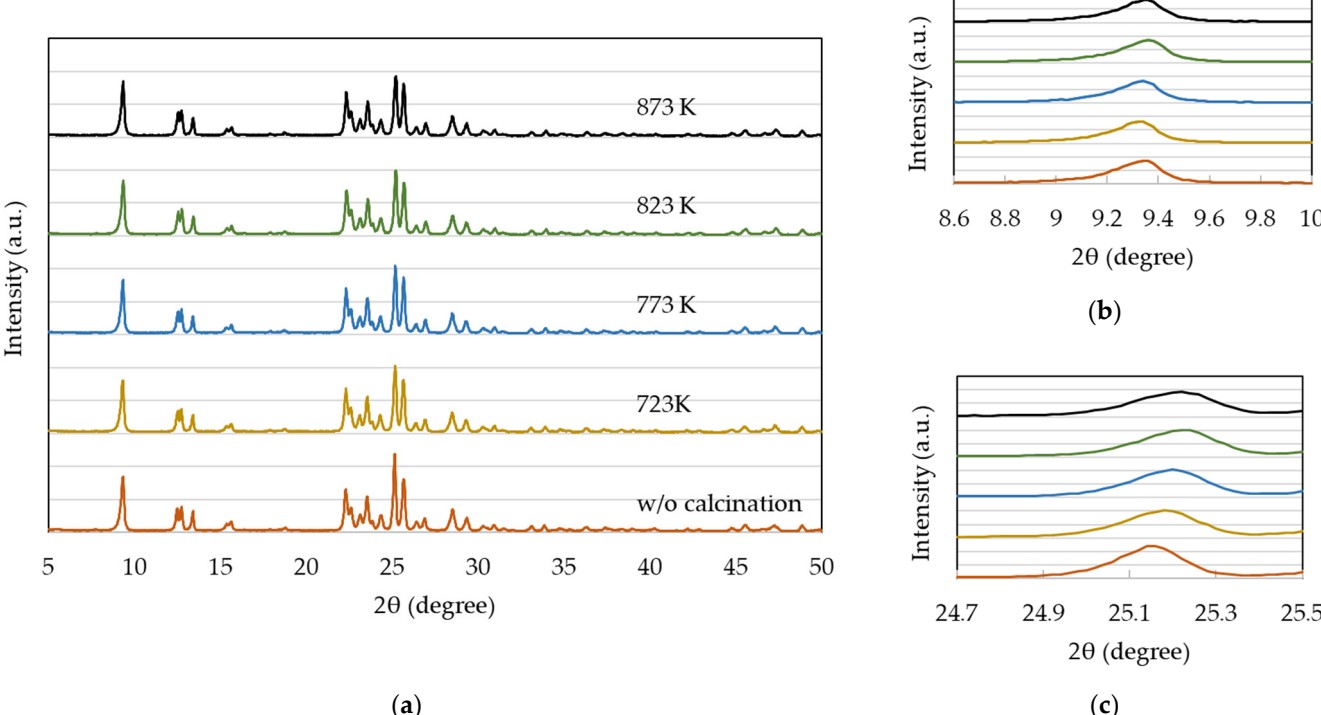

**Figure 5.** XRD patterns of 720NHA (FER) at different calcination temperatures: (**a**) 5–50°; (**b**) 8.6–10°; and (**c**) 24.7–25.5°.

With an increase in the calcination temperature, the peak at 25.1° shifted to a higher 2θ value while slightly decreasing in intensity (Figure 5c). This indicates that the structures of the pores and crystals changed due to calcination. Such peak shifts and intensity decreases reflect a decrease in the lattice constant and degradation in crystallinity, respectively. In contrast, an increase in the temperature caused no significant changes in the surface area and pore volume (Table 3). Considering the correlation of higher calcination temperatures to an increase in the weight of 720NHA (FER) (Figure 2), these results suggested that the incorporated oxygen led not only to crystal structure degradation, but also to a decrease in the interlayer distance.

For 840NHA (MFI), a peak characteristic of the MFI structure was observed; however, it was not affected by changes in the calcination temperature (Figure S3). A peak characteristic of the BEA structure was observed for 940HOA (BEA) (Figure S4). Calcination led to increases in the peak intensities at 7.9° and 22.5°, indicating the promotion of crystallization. In contrast, the XRD patterns of the calcined 940HOA (BEA) were independent of the calcination temperature (Figure S4).

The $NH_3$-TPD for 720NHA (FER), 840NHA (MFI), and 940HOA (BEA) at different calcination temperatures are shown in Figure 6. The acidity determined from $NH_3$-TPD measurements is useful for comparing the acidic properties of the catalysts, but these can be generally overestimated [28]. It is not possible to distinguish between the originally contained $NH_3$ and the newly adsorbed $NH_3$ due to heating because the parent 720NHA (FER) and 840NHA (MFI) zeolites contain $NH_3$. Therefore, $NH_3$-TPD was not performed using the zeolites without calcination.

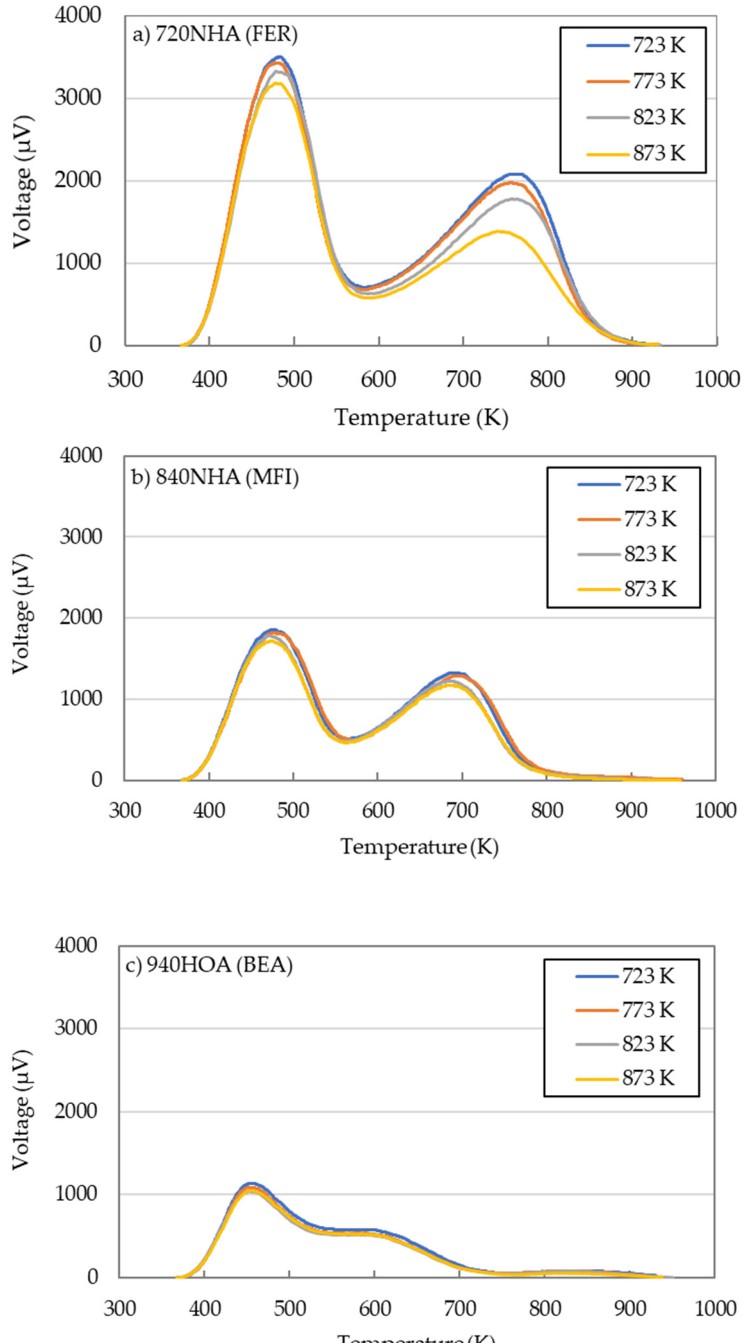

**Figure 6.** NH$_3$-TPD profiles for three zeolites (**a**) 720NHA (FER), (**b**) 840NHA (MFI), (**c**) 940HOA (BEA) at different calcination temperatures.

720NHA (FER) and 840NHA (MFI) showed two relatively clear peaks (Figure 6a,b), while 940HOA (BEA) exhibited three broad peaks (Figure 6c). The NH$_3$-TPD profiles cannot strictly distinguish between the Lewis and Brønsted acidity. Therefore, NH$_3$ desorbed at 373–573 K and above 573 K is considered to be desorption from weak acid sites and strong acid sites, respectively [36]. Here, the Brønsted acid sites are contained in the strong acid sites.

The effects of the calcination temperature on the acidic properties are included in Table 4. The number of acid sites and acid strength mainly depended on the zeolite species and calcination temperature. For 720NHA (FER), when the calcination temperature was 773 K, both acid sites showed the highest acid amount. In contrast, for 840NHA (MFI) and 940HOA (BEA), both acid sites decreased monotonically with an increase in the calcination

temperature. These results could be attributed to a decrease in the Brønsted acid sites because an increase in the temperature promoted dehydration [37,38].

**Table 4.** Acidic properties of 720NHA (FER), 840NHA (MFI), and 940HOA (BEA) at different calcination temperatures.

| Material | Topology | Calcination Temp. (K) | Weak [1] (mmol/g) | Strong [2] (mmol/g) | Total (mmol/g) |
|---|---|---|---|---|---|
| 720NHA | FER | 723 | 1.358 | 1.369 | 2.727 |
| | | 773 | 1.406 | 1.373 | 2.779 |
| | | 823 | 1.386 | 1.288 | 2.674 |
| | | 873 | 1.297 | 0.999 | 2.296 |
| 840NHA | MFI | 723 | 0.774 | 0.749 | 1.523 |
| | | 773 | 0.774 | 0.747 | 1.521 |
| | | 823 | 0.760 | 0.713 | 1.473 |
| | | 873 | 0.735 | 0.675 | 1.410 |
| 940HOA | BEA | 723 | 0.498 | 0.268 | 0.766 |
| | | 773 | 0.486 | 0.255 | 0.741 |
| | | 823 | 0.463 | 0.243 | 0.706 |
| | | 873 | 0.469 | 0.231 | 0.700 |

[1] calculated based on desorbed $NH_3$ amounts at 373–573 K; [2] calculated based on desorbed $NH_3$ amounts above 573 K.

Ferrierite zeolites contain a two-dimensional pore structure in which 10 MR (member ring, 5.4 × 4.2 Å) channels and 8 MR (4.8 × 3.5 Å) channels intersect. The kinetic diameters for $NH_3$ and $N_2$ were 2.6 and 3.64 Å, respectively [39]. The $NH_4^+$ functional group would have smaller length than the kinetic diameter; however, it may prevent $N_2$ from diffusing into the micropores. Calcination at 723 K resulted in the desorption of $NH_3$ and led to the formation of micropores. For all three zeolites, an increase in the temperature above 723 K promoted dehydration, resulting in slight effects on the micropore surface area and micropore volume.

*2.3. Correlation of Observed Values to n-Butene Yield*

The optimal calcination temperature was identified for 720NHA (FER) (Table 2). Determination of the combination of parameters that are highly correlated to the *n*-butene yield is important not only for improving the catalytic performance in the DTO reaction, but also for enhancing the economics of the 1,3-BD production process shown in Figure 1.

Multiple regression analysis, which can provide effective information for the development of catalysts, was performed to model the performance of the catalysts [40–42]. Malik et al. reported that the analysis results can explain the different variation trends of catalytic activities using similar catalysts [42].

In this study, six observed values including micropore surface area ($x_1$), external surface area ($x_2$), total pore volume ($x_3$), micropore volume ($x_4$), weak acid ($x_5$), and strong acid ($x_6$) were considered as explanatory variables, and the correlation with *n*-butene yield (y) was examined.

The six explanatory variables, *n*-butene yield, and correlation coefficients ($r_{ij}$) between the explanatory variables were organized (Table S1). The tolerance ($1 - r_{ij}^2$) for the selected explanatory variables is presented in Table 5. When the tolerance is less than the critical value (0.1), multicollinearity is observed, resulting in a decrease in the correlation accuracy. Therefore, combinations of explanatory variables other than $x_1 - x_4$, $x_2 - x_3$, and $x_5 - x_6$ were employed for multiple regression analysis.

The absolute values of the correlation coefficients with *n*-butene yield (y) were in the order of $x_6$ (0.836) > $x_5$ (0.766) > $x_4$ (0.701) > $x_1$ (0.676) > $x_2$ (0.310) > $x_3$ (0.156) (Table S1). As $x_2$ and $x_3$ had small correlation coefficients of ≤0.31, these were not used as explanatory variables.

**Table 5.** Tolerance between selected explanatory variables.

| Selected Explanatory Variables | Tolerance |
|:---:|:---:|
| $x_1 - x_2$ | 0.981 |
| $x_1 - x_3$ | 0.984 |
| $x_1 - x_4$ | 0.014 |
| $x_1 - x_5$ | 0.301 |
| $x_1 - x_6$ | 0.175 |
| $x_2 - x_3$ | 0.092 |
| $x_2 - x_4$ | 0.949 |
| $x_2 - x_5$ | 0.618 |
| $x_2 - x_6$ | 0.784 |
| $x_3 - x_4$ | 0.998 |
| $x_3 - x_5$ | 0.844 |
| $x_3 - x_6$ | 0.948 |
| $x_4 - x_5$ | 0.213 |
| $x_4 - x_6$ | 0.117 |
| $x_5 - x_6$ | 0.052 |

The explanatory variables, residual sum of squares (RSS), F-ratio, regression equation, and contribution rate are included in Table 6. The RSS is derived from the error between the data and regression equation, and a small RSS is favored. The F-ratio represents the decrease rate of the RSS when the explanatory variables are added, and if it is two or more, it indicates that the addition of the explanatory variables is effective.

**Table 6.** Multiple regression analysis data.

| Trial | Explanatory Variable | Residual Sum of Squares (RSS) | F-Ratio | Regression Equation | Contribution Rate (%) |
|:---:|:---:|:---:|:---:|:---:|:---:|
| 1 | None | 2894 | | | |
| 2 | $x_6$ | 253 | 104.40 | $y = 0.74 + 16.61x_6$ | 69.9 |
| 3 | $x_6, x_4$ | 201 | 2.34 | $y = -30.04 + 113.43x_4 + 30.22x_6$ | 76.1 |
| 4 | $x_6, x_1$ | 219 | 1.39 | $y = -20.46 + 25.26x_6 + 0.03x_1$ | 74.0 |

Without the explanatory variables (Trial 1), the RSS is the sum of the squares of the *n*-butene yield (2894). When $x_6$ with the highest correlation coefficient was added to y (Trial 2), the RSS decreased to 253 and the F-ratio was $\geq 2$; therefore, $x_6$ was adopted as the explanatory variable.

Although $x_5$ had the second highest correlation coefficient, no additional consideration was given because of the low tolerance of $x_5 - x_6$ (Table 5).

The addition of $x_4$, which had the third highest correlation coefficient, decreased the RSS to 201 and the F-ratio was >2 (Trial 3). Therefore, $x_4$ was also used as an explanatory variable. In contrast, the addition of $x_1$ (Trial 4) resulted in an F-ratio of <2; therefore, it was not adopted.

After Trial 3, the addition of $x_1$ was not performed because of the low tolerance of $x_1 - x_4$ (Table 5), and multiple regression analysis was completed. When $x_4$ (micropore volume) and $x_6$ (strong acid) were used as the explanatory variables (Trial 3), the contribution of the regression equation was 76.1%, indicating that the *n*-butene yield could be better explained by 6.2% in comparison to that in Trial 2.

The dual-cycle mechanism has been proposed as a DTO reaction mechanism (Figure 7) [43–45]. Assuming that this mechanism is followed, it is necessary to promote propylene methylation and suppress *n*-butene methylation to increase *n*-butene yield in the olefin cycle.

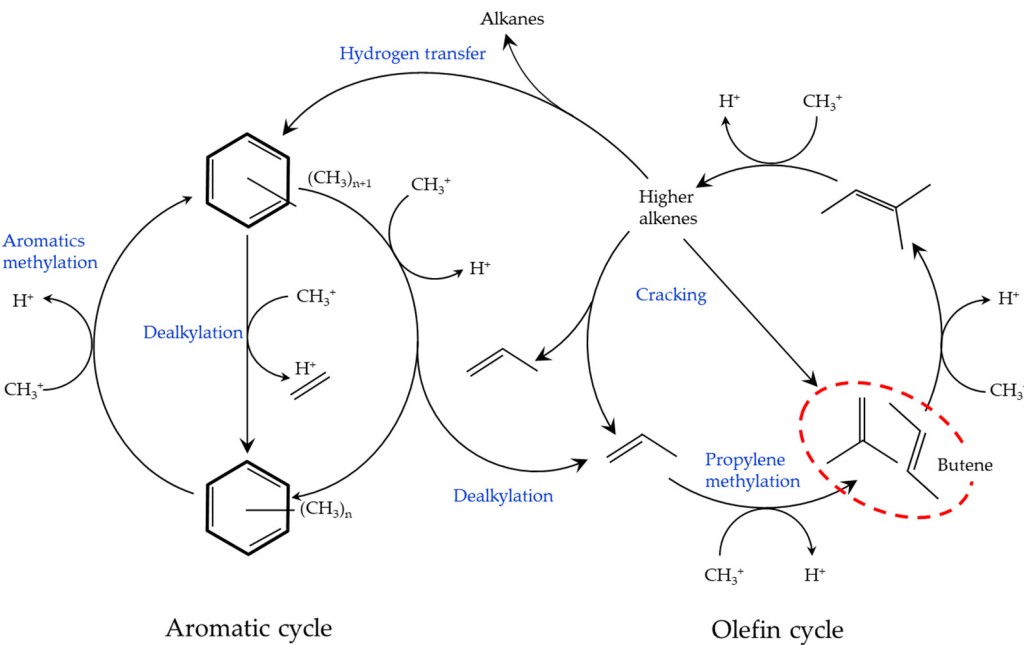

**Figure 7.** Dual-cycle mechanism. Adapted with permission from Ref. [43]. Copyright 2012, Elsevier. Adapted with permission from Ref. [44]. Copyright 2012 Wiley. Adapted with permission from Ref. [45]. Copyright 2012 American Chemical Society.

At the initial stage of the DTO reaction, a hydrocarbon pool is formed. In the MTH reaction, higher alkene selectivity requires active methyl benzenes, which are important intermediates as the hydrocarbon pool, followed by the production of ethylene and propylene via dealkylation [44]. In this study, the active methyl benzene is produced in the aromatic cycle; however, the propylene is predominantly derived from the olefin cycle, because the activation energy of the methylation reaction by DME is comparable to, or in excess of, that of methanol [46].

Propylene is required to promote the olefin cycle. The propylene-to-ethylene ratio is important for high alkene selectivity in the MTH reaction [44]. The ratio in this study ($C_3^=/C_2^=$) decreased in the order of 720NHA (2.2) > 840NHA (2.0) > 940HOA (1.3) (Table 2), which is consistent with the order of the *n*-butene yield.

Here, it is necessary to consider the selective synthesis of butene. Brønsted acid sites, which are the active sites of alkene methylation, are present in strong acids. The increase in strong acid sites is favorable for the methylation of butene and propylene. Menges et al. reported that the rate of butene production by methylation was 1.7 times faster than the rate of butene disappearance [47]. Ortega et al. reported that butene was relatively stable because of the high activation energy of the butene disappearance rate [48]. As 720 NHA (FER) had a large amount of strong acid sites, which acted as active alkylation sites, the yields of hydrocarbons with many carbon atoms were higher than those with 840NHA (MFI) and 940HOA (BEA) (Table 2).

Zhao et al. reported that the cage size was highly correlated to the light olefin yield, and the increase in the size favored the production of propylene and butene in the DTO reaction using AEI/CHA zeolites [10]. Finally, *n*-butene was released from the zeolite pores; therefore, a micropore volume that can provide an appropriate residence time is required.

The correlation coefficient between the micropore volume and *n*-butene yield was negative (Table S1). Small micropore volume is desirable; however, micropores with strong acid sites are required. This speculation corresponds to the results of the multiple regression analysis.

Hill et al. reported that the reaction rate constant for *iso*-butene alkylation was 40 times higher than those for 1-butene, *trans*-2-butene, and *cis*-2-butene [49]. This result indicates

that in the present study, it is necessary to suppress the generation of *iso*-butene during propylene alkylation. Khitev et al. reported an increase in the selectivity of *iso*-butene from 1-butene with an increase in the ratio of mesopores to micropores in the ferrierite zeolite [50]. In contrast, it supported the speculation that micropores that caused steric hindrance were advantageous for suppressing skeletal isomerization.

Suh et al. reported the selective *iso*-butene synthesis from 1-butene using the zirconia with a mean pore diameter of ca. 5 Å [51]. This result suggested that the pore diameter was also an important factor, and that a diameter of 5 Å or less was required to suppress *iso*-butene production from *n*-butene.

In this study, 940HOA (BEA) showed high *iso*-butane and *n*-butane yields (Table 2). The amount of strong acid sites containing Brønsted acid sites for alkene production was smaller than those in 720NHA (FER) and 840NHA (MFI) (Table 4). Laredo et al. reported an *iso*-paraffin selectivity of ≥70% in the propylene addition reaction to benzene using a beta zeolite [52]. A high proportion of Lewis acid sites to Brønsted acid sites promoted paraffin production. This was consistent with the high paraffin yield of 940HOA (BEA), which had relatively fewer strong acid sites.

Olsbye et al. mentioned that in the MTH reaction, the pore and/or cavity size affects the production of important intermediates and the ethylene to propylene ratio [44]. In particular, the cavity size is closely related to the micropore volume. Accordingly, considering the multiple regression analysis, it was speculated that 720NHA (FER) had micropores in which the active sites for alkene alkylation were dispersed and its volume suppressed *iso*-butene formation spatially and provided the desired residence time.

Ferrierite zeolites have a problem in catalyst life because coke is likely to be deposited. Future studies will focus on extending the lives of the promising ferrierite zeolites identified in this study.

## 3. Materials and Methods

### 3.1. Catalyst Preparation

Ten types of commercially available zeolites manufactured by Tosoh Co., Ltd. (Tokyo, Japan) were used, including two types of FER zeolite (720NHA, 720KOA), five types of MFI zeolite (820NHA, 840NHA, 822HOA, 840HOA, 890HOA), BER zeolite (940HOA), MOR zeolite (640HOA), and FAU zeolite (385HUA). The zeolite species, topology, cation type, crystal size, and particle size are listed in Table 7. As a pretreatment method, calcination was performed. After attaining the desired temperature (from room temperature) in one hour, it was maintained for an additional two hours.

**Table 7.** Topology, cation type, crystalline size, and particle size of various zeolites.

| Material | Topology | Cation Type | Crystalline Size [1] (μm) | Particle Size [1] (μm) |
|---|---|---|---|---|
| 720NHA | FER | $NH_4^+$ | ≤1 | 6 |
| 720KOA | | $K^+$ | ≤1 | 20 |
| 820NHA | | $NH_4^+$ | 0.1 × 0.5 | 5 |
| 840NHA | | $NH_4^+$ | 2 × 4 | 10 |
| 822HOA | MFI | $H^+$ | 0.1 × 0.5 | 5 |
| 840HOA | | $H^+$ | 2 × 4 | 10 |
| 890HOA | | $H^+$ | 2 × 5 | 10 |
| 940HOA | BEA | $H^+$ | 0.5–1 | 4 |
| 640HOA | MOR | $H^+$ | 0.1 × 0.5 | 12 |
| 385HUA | FAU | $H^+$ | 0.7–1.0 | 2–3 |

[1] referring to manufacturer catalog.

### 3.2. Catalyst Characterization

The weight change of the zeolites under calcination conditions was observed using a thermogravimetric analyzer (Thermo plus TG8120, Rigaku, Tokyo, Japan). Approximately 10 mg of zeolite was placed in a platinum crucible and heated from room temperature up to 973 K at a rate of 5 K/min under 60 mL/min air flow at atmospheric pressure.

The specific surface area and porosity were measured using a specific surface area/pore distribution measurement apparatus (Belsorp-mini II, MicrotracBEL Corp., Osaka, Japan). The external surface area, micropore volume, and total pore volume were calculated using the t-plot method, where P and $P_0$ are the actual and saturated $N_2$ pressures, respectively.

The crystallinity of the zeolites was determined using a powder X-ray diffractometer (RINT TTR III, Rigaku, Japan) equipped with Cu K$\alpha$ radiation. XRD measurements were recorded in the $5° < 2\theta < 50°$ angle range with a voltage of 50 kV and a current of 300 mA. The step width and scan speed were $0.02°$ and $2°$/min, respectively.

$NH_3$-TPD was performed by employing a BELECAT-B (MicrotracBEL Corp., Osaka, Japan) equipped with a thermal conductivity detector (TCD). For each run, approximately 0.05 g of zeolite was heated to 723 K at 10 K/min under a 50 mL/min He flow and then retained for one hour. After cooling to 373 K and holding for 10 min, the $NH_3$/He gas mixture (5/95 vol %) was supplied at 50 mL/min for 30 min. Thereafter, it was heated to 973 K at a heating rate of 10 K/min under a 50 mL/min He flow, and desorbed $NH_3$ was detected using TCD.

### 3.3. DTO Reaction

A schematic of the experimental apparatus used for the DTO reaction is shown in Figure 8. The catalytic reaction was carried out under normal pressure using a fixed-bed reactor. Compared to the fluidized bed DTO reactor [53], temperature distribution may be observed in the catalyst layer, but the fixed-bed reactor was selected because of its ease of operation. Zeolite (0.2 g) was fixed in a quartz reaction tube (internal diameter (id): 9.0 mm; length: 235 mm) using quartz wool. The tip of the thermocouple was installed near the center of the catalyst layer to measure the actual temperature of the zeolite.

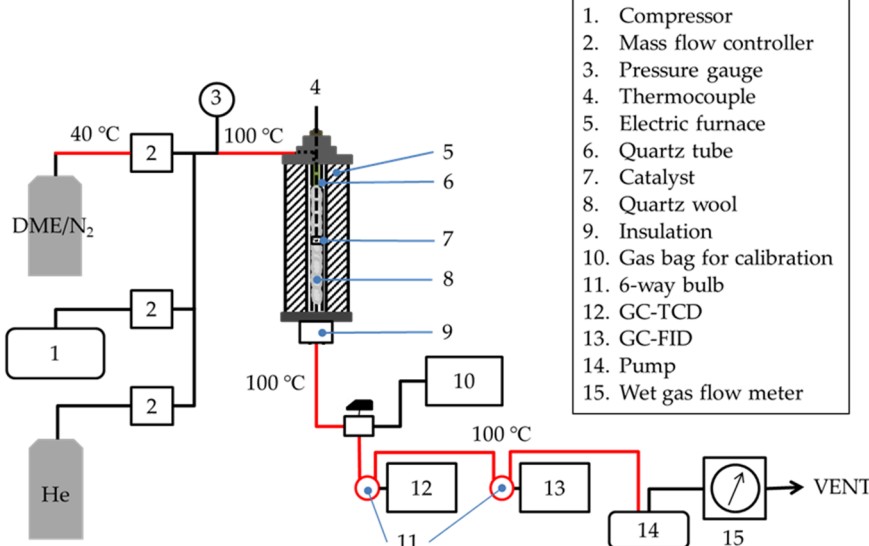

**Figure 8.** Schematic of the experimental apparatus for DTO reaction.

The catalyst layer was heated using an electric furnace, while air (200 mL (STP)/min) was supplied to the reactor using a compressor. After the catalyst layer temperature reached approximately 623 K, the air was switched to a DME/$N_2$ gas mixture (10/90 vol %, 200 mL (STP)/min) and supplied for five minutes. Owing to the promotion of the reaction, the catalyst layer temperature increased to approximately 673 K. Thereafter, the DME/$N_2$

gas mixture was set to 49 mL (STP)/min, and the reaction was considered as having started. Then, the catalyst layer temperature was adjusted to $673 \pm 5$ K.

Ten minutes after the start of the reaction, the product gas leaving the reactor was analyzed online using gas chromatography (GC)-TCD (GC-2014, Shimadzu, Kyoto, Japan) and GC coupled to a flame ionization detector (GC-FID; GC-2014, Shimadzu, Kyoto, Japan). GC-TCD with a MICROPACKED-ST column (id: 3.0 mm; length: 6 m; Shinwa Chemical Industries Ltd., Kyoto, Japan), GC-FID with a DB-5 capillary column (id: 0.53 mm; length: 60 m; thickness of the stationary phase: 5 μm; Agilent Technology, Tokyo, Japan), and HP-PLOT/Q (id: 0.53 mm; length: 30 m; thickness of the stationary phase: 40 μm; Agilent Technology, Tokyo, Japan) were connected in series.

The GC-FID heating program included temperature holding at 323 K for two minutes, an increase to 363 K at 8 K/min, and holding for 16 min. Thereafter, the temperature was increased to 371 K at 1 K/min and 453 K at 8 K/min and held for 60 min.

The DME conversion and product yield were calculated as follows.

$$\text{DME conversion [\%]} = (\text{DME}_{in} \text{ [mol]} - \text{DME}_{out} \text{ [mol]})/\text{DME}_{in} \text{ [mol]} \times 100 \qquad (1)$$

$$\text{Product yield [C-mol\%]} = \text{Carbon number in product gas leaving the reactor [C-mol]} / \text{Total carbon number in feed gas [C-mol]} \times 100 \qquad (2)$$

For products with up to four carbon atoms, qualitative analysis was clearly possible using the GC-FID data. Therefore, for hydrocarbons with up to four carbon atoms, the area ratio to nitrogen was calibrated and used to calculate the DME conversion and product yield. The effective carbon number of MeOH was set to 0.4.

Additionally, many peaks belonging to compounds with five or more carbon atoms were observed. The peaks with retention times of 36–42, 42–50, and 50–62 min were speculated to correspond to the hydrocarbons with five, six, and seven carbon atoms, respectively. In some cases, peaks were also observed at 88–98 min. Due to the unavailability of the GC-mass spectrometry system, qualitative analysis could not be performed. Therefore, the product yield was calculated by considering benzene as the unknown. The effective carbon numbers of the hydrocarbons with five, six, and seven carbon atoms were five, six, and seven, respectively, and the effective carbon number of benzene was six, which was used for the calculation of the product yield.

## 4. Conclusions

In the present study, zeolite catalysts that can efficiently afford *n*-butene from DME are identified. The conclusions are listed below.

1.  A screening study indicated that 720NHA (FER) showed the highest *n*-butene yield.
2.  The effects of calcination temperature on the catalytic performance in the DTO reaction were investigated for 720NHA (FER), 840NHA (MFI), and 940HOA (BEA). The 720NHA (FER) provided the highest *n*-butene yield at a calcination temperature of 773 K. In contrast, for 840NHA (MFI) and 940HOA (BEA), the *n*-butene yield decreased with an increase in the calcination temperature.
3.  Multiple regression analysis was performed on the six observed values and *n*-butene yield. The strong acid site and micropore volume were selected as statistically important explanatory variables. Strong acid sites should be dispersed, and the spatially narrow micropores should have a volume that provides an appropriate residence time in zeolites to suppress *iso*-butene production.

**Supplementary Materials:** The following are available online at https://www.mdpi.com/article/10.3390/catal11060743/s1, Figure S1: N$_2$ adsorption–desorption isotherms for 840NHA at 77 K (MFI); Figure S2: N$_2$ adsorption-desorption isotherms for 940HOA at 77 K (BEA); Figure S3: XRD patterns of 840NHA (MFI) at different calcination temperatures; Figure S4: XRD patterns of 940HOA (BEA) at different calcination temperatures; Table S1: Supplementary table for multiple regression analysis.

**Author Contributions:** Data curation, T.H.; writing—original draft, T.H.; writing—review and editing, M.A. and Y.E.; project supervision, Y.E. The manuscript has been read and revised by all the authors before submission. All authors have read and agreed to the published version of the manuscript.

**Funding:** This work was supported by the Fundamental Research Fund of the National Institute of Advanced Industrial Science and Technology (AIST), Japan, and did not receive any specific grant from funding agencies in the public, commercial, or not-for-profit sectors.

**Data Availability Statement:** Not applicable.

**Acknowledgments:** The authors are grateful for the fundamental grant from AIST. The authors are also thankful for valuable comments from Shinji Fujimoto, Hirohmi Watanabe, and Shotaro Ito, and experimental assistance from Yuna Kira and Konoha Shiimori.

**Conflicts of Interest:** The authors declare no conflict of interest.

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
