# Peer review of "n-Butene Synthesis in the Dimethyl Ether-to-Olefin Reaction over Zeolites"

_catalysts, doi:10.3390/catal11060743_

Round 1

Reviewer 1 Report

The manuscript by Hanaoka et al. focuses on catalyzing dimethyl ether to olefin (DTO) reaction, one of the key steps in
converting biomass and syngas to light olefins.They screened 10 zeolite candidates, and studied the effect of pretreatment
temperatures and active site acidity on the conversion efficiency to n-butene, and further tried identifying the descriptors. 
Overall, the conclusions are supported by the experimental results. However, there are some comments that the authors need to consider
in the revision.

1. Cations. The authors compared the catalytic activities of FER zeolite with NH4+ (720NHA) and K+ (720KOA) as the cation, and 
observed much better performance in the former due to the presence of acid after NH3 desorption. Is it possible to start with 
the protonic form of the FER zeolite? This might further increase the Bronsted acid sites, but keep the same micropore volume. 

2. Page 3. The authors wrote"Generally, the acid strength increased, and the acid amount decreased with increasing the SiO2/Al2O3
molar ratio". It is a bit confusing to me what the acid strength means here. Does it refer to the pKa of the acidic sites? 
and why would the acid strength increase when you decrease the percentage of the Lewis acid, i.e. Al2O3? 

3. Page 3. The authors wrote "Magomedova et al reported that MeOH selectivity increased and ethylene selectivity decreased with an 
increase in the zeolite acid strength [25]". However, the cited reference concluded "changes in product selectivity for HZSM-5/Al2O3,
-Zr, and Mg-modified samples were similar and did not depend on catalyst acidity". (ref. 25, Conclusions) The authors might 
want to reconcile this difference. 

4. What atmosphere was the TG experiments done at? There is no detail about TG in the experimental section. Also, it would also be useful
if the authors can explain a bit more on the TG curves. Say, what does the weight loss before 600 K correspond to. 

5. Follow-up on the TG curves. The authors wrote "The TG results suggested that calcination above the DTO reaction temperature (673K)
led to variations in the catalytic performance". This sentence is inappropriate here, because no catalysis has been done yet.

6. Do the authors know what is the structural feature corresponding to the 25.1 degree in the XRD pattern? This seems to be a key 
structural change in the 720NHA zeolite induced by the calcination process. Some further discussions, or citations here are useful.

7. What is the approximate size of the micropores? It is hard to imagine NH4+ would be able to significantly block pores unless 
they are comparable in size. 

8. Fig. 7 cited the dual-cycle mechanism, but there is no mentioning of the aromatic cycle in the text. The authors can add more
explanations in the main text or in the caption. 

Author Response

Dear Reviewer 1,

We appreciate your comments about our manuscript and are delighted with your favorable response. We have revised the manuscript in accordance with your recommendations and hope that the changes that have been made are to your satisfaction.

Comments and Suggestions from Reviewer 1

The manuscript by Hanaoka et al. focuses on catalyzing dimethyl ether to olefin (DTO) reaction, one of the key steps in converting biomass and syngas to light olefins. They screened 10 zeolite candidates, and studied the effect of pretreatment temperatures and active site acidity on the conversion efficiency to n-butene, and further tried identifying the descriptors. Overall, the conclusions are supported by the experimental results. However, there are some comments that the authors need to consider in the revision.

Comment #1

Cations. The authors compared the catalytic activities of FER zeolite with NH4+ (720NHA) and K+ (720KOA) as the cation, and observed much better performance in the former due to the presence of acid after NH3 desorption. Is it possible to start with the protonic form of the FER zeolite? This might further increase the Bronsted acid sites, but keep the same micropore volume.

Response #1

Thank you for your suggestion. Unfortunately, the protonic form of 720NHA is not commercially available. However, it is possible to prepare FER zeolites by extending the appropriate calcination time. Currently, we are working on improving the catalytic performance of ferrierite zeolites. The effect of calcination time will be investigated for zeolites with maintained micropore volumes.

Comment #2

Page 3. The authors wrote "Generally, the acid strength increased, and the acid amount decreased with increasing the SiO2/Al2O3 molar ratio". It is a bit confusing to me what the acid strength means here. Does it refer to the pKa of the acidic sites? and why would the acid strength increase when you decrease the percentage of the Lewis acid, i.e. Al2O3?

Response #2

Thank you for your valuable comment. We misunderstood the relationship between the acid strength and the SiO2/Al2O3 molar ratio. This sentence was inappropriate and has been deleted from the Introduction.

Comment #3

Page 3. The authors wrote "Magomedova et al reported that MeOH selectivity increased and ethylene selectivity decreased with an increase in the zeolite acid strength [25]". However, the cited reference concluded "changes in product selectivity for HZSM-5/Al2O3,-Zr, and Mg-modified samples were similar and did not depend on catalyst acidity". (ref. 25, Conclusions) The authors might want to reconcile this difference.

Response #3

Thank you for your valuable comment. As the reviewer indicated, the product selectivity was independent of the catalyst acidity in Ref. 25, and our assertion is not appropriate. Therefore, this sentence has been deleted. As an alternative, Ref. 25 in the original manuscript was used to discuss the reaction mechanism as Ref. 28 in the revised manuscript. Moreover, we reviewed the literature on the hydration of DME using MFI zeolite. A high MeOH selectivity was reported at 473–593 K, but no MeOH production was observed at 673 K owing to thermodynamic disadvantages. Although not the focus of this study, it is an interesting result, and we will consider this high MeOH selectivity in the future. This explanation has been added to lines 4–10 in Page 3.

Comment #4

What atmosphere was the TG experiments done at? There is no detail about TG in the experimental section. Also, it would also be useful if the authors can explain a bit more on the TG curves. Say, what does the weight loss before 600 K correspond to.

Response #4

In lines 1–4 in the Section 3.2. (Page 13), the measurement conditions using the thermogravimetric analyzer were added. In lines 7–10 in the Section 2.2. (Page 3), an explanation regarding the weight loss before 600 K was added.

Comment #5

Follow-up on the TG curves. The authors wrote "The TG results suggested that calcination above the DTO reaction temperature (673K) led to variations in the catalytic performance". This sentence is inappropriate here because no catalysis has been done yet.

Response #5

Thank you for your comment. In lines 13–15 in the Section 2.2. (Page 3), the sentence was replaced with “The TG results suggested that calcination above the DTO reaction temperature (673 K) caused variations in the structural parameters, crystallinity, and acidic properties of the zeolites.”

Comment #6

Do the authors know what is the structural feature corresponding to the 25.1 degree in the XRD pattern? This seems to be a key structural change in the 720NHA zeolite induced by the calcination process. Some further discussions, or citations here are useful.

Response #6

Thank you for your comment. As the reviewer notes, the only peak shift at 25.1° does seem crucial. We researched the literature; unfortunately, no relevant information could be found. Therefore, an explanation regarding the expected structural change was added to lines 3–9 in the third paragraph (Page 7).

Comment #7

What is the approximate size of the micropores? It is hard to imagine NH4+ would be able to significantly block pores unless they are comparable in size.

Response #7

Ferrierite zeolites contain two-dimensional pore structures in which 10-MR (5.4 × 4.2 Å) channels and 8-MR (4.8 × 3.5 Å) channels intersect. The kinetic diameters for ammonia and N2 are 2.6 and 3.64 Å, respectively. In this experiment, the NH4+ molecules adsorbed on the inner walls of the micropores possibly prevented N2 from diffusing into the micropores. In lines 3–7 in the last paragraph (Page 8), an explanation has been added. In this study, the pore size was not used for multiple regression analysis, but it was considered to be an important factor. We have added a relevant reference and incorporated a description of the appropriate size in the last paragraph in Page 12.

Comment #8

Fig. 7 cited the dual-cycle mechanism, but there is no mentioning of the aromatic cycle in the text. The authors can add more explanations in the main text or in the caption.

Response #8

Thank you for your valuable comment. It is important to consider the aromatic cycle for a deeper discussion of the DTO reaction mechanism. Therefore, Refs. 44 and 46 regarding the reaction mechanism have been added. The explanation regarding the expected reaction mechanism was added to the last two paragraphs in Page 11, considering both the DTO and MTH reaction mechanisms.

Reviewer 2 Report

The paper begins with stating the significance of 1,3-butadiene as a chemical feedstock and how current industrial trends require an alternative route but the explanation is lacking. I think the statement "The utilization of 1,3-BD can improve the quality of these materials," is too vague and general to be useful to the reader. The introduction is adequate, if not brief, but it would be better if the conclusion referred back to the introduction better, since the initial justification for carrying out this study is not mentioned again afterwards.

The written English has been carefully spell checked, with almost no grammatical errors. However, I do have an issue with the abundance of short sentences and repetitive phrases/words. This makes reading the script quite arduous at times and disrupts the flow of the text significantly, especially in the introduction where clear prose is very important.

Here is a good example of 4 consecutive short sentences which could easily be linked (not necessarily all together):

"Currently, 1,3-BD is mainly produced as a co-product in the production of ethylene from naphtha. In recent years, the amount of ethylene produced from shale gas, which is cheaper than naphtha, has increased. As it is difficult to produce 1,3-BD from shale gas, there are concerns about the adequate supply of 1,3-BD. Therefore, 1,3-BD synthesis from biomass is desirable and has attracted considerable attention [2,3]."

Is the evidence for shale gas being more difficult than naphtha for producing 1,3-butadiene in references 2 + 3 also? Words taken from Latin and other languages should be in italics.

I find it odd that despite showing the dual-cycle mechanism there is little reference to it and no mention of any benzene or alkyl benzene products forming from the further reaction of C3/C4 products. Upon reading in other publications, it seems like the formation of methyl benzene intermediates (the hydrocarbon pool) is integral to the formation of light olefins, which are less reactive than the DME feed.

I think that the regression analysis needs to refer back to some of the work done by others to give it more context.

Whilst I agree that there has not been a study to specifically focus on n-butenes, there have been other studies looking at the distribution of products from the reaction and insight into this, indeed from this journal ("Dimethyl Ether to Olefins over Modified ZSM-5 Based Catalysts Stabilized by Hydrothermal Treatment" https://doi.org/10.3390/catal9050485)

Overall, this is a carefully carried out and written-up study of a series of commercially available zeolites but it lacks mechanistic depth and explanation of the results. Given the solid experimental foundation of the study, I think that it could be accepted with major revisions to the discussion of the results and without carrying out any additional experiments. There are some comprehensive reviews, for example: ("Conversion of Methanol to Hydrocarbons: How Zeolite Cavity and Pore Size Controls Product Selectivity." https://doi.org/10.1002/anie.201103657.), which give excellent background on some of the discussions necessary to help explain the results.

Author Response

Dear Reviewer 2,

We appreciate your comments about our manuscript and are delighted with your favorable response. We have revised the manuscript in accordance with your recommendations and hope that the changes that have been made are to your satisfaction.

Comments and Suggestions from Reviewer 2

Comment #1

The paper begins with stating the significance of 1,3-butadiene as a chemical feedstock and how current industrial trends require an alternative route, but the explanation is lacking. I think the statement "The utilization of 1,3-BD can improve the quality of these materials," is too vague and general to be useful to the reader. The introduction is adequate, if not brief, but it would be better if the conclusion referred back to the introduction better, since the initial justification for carrying out this study is not mentioned again afterwards.

The written English has been carefully spell checked, with almost no grammatical errors. However, I do have an issue with the abundance of short sentences and repetitive phrases/words. This makes reading the script quite arduous at times and disrupts the flow of the text significantly, especially in the introduction where clear prose is very important.

Response #1

Thank you for your comment. First, the importance of 1,3-BD addition to rubber and plastic was added to lines 3–5 in the Introduction. A brief summary was added to focus on the objective and results in lines 5–8 in the third paragraph in Page 2. Moreover, we used an English proofreading service for all manuscripts to reduce repetitive expressions.

Comment #2

Here is a good example of 4 consecutive short sentences which could easily be linked (not necessarily all together):

"Currently, 1,3-BD is mainly produced as a co-product in the production of ethylene from naphtha. In recent years, the amount of ethylene produced from shale gas, which is cheaper than naphtha, has increased. As it is difficult to produce 1,3-BD from shale gas, there are concerns about the adequate supply of 1,3-BD. Therefore, 1,3-BD synthesis from biomass is desirable and has attracted considerable attention [2,3]."

Is the evidence for shale gas being more difficult than naphtha for producing 1,3-butadiene in references 2 + 3 also? Words taken from Latin and other languages should be in italics.

Response #2

Thank you for your comment. This study focused on the development of a DTO catalyst, and references 2 and 3 are related to the importance of 1,3-BD production from biomass. Therefore, to focus on the objective, the sentences you noted were deleted from the Introduction.

Comment #3

I find it odd that despite showing the dual-cycle mechanism, there is little reference to it and no mention of any benzene or alkyl benzene products forming from the further reaction of C3/C4 products. Upon reading in other publications, it seems like the formation of methyl benzene intermediates (the hydrocarbon pool) is integral to the formation of light olefins, which are less reactive than the DME feed.

Whilst I agree that there has not been a study to specifically focus on n-butenes, there have been other studies looking at the distribution of products from the reaction and insight into this, indeed from this journal ("Dimethyl Ether to Olefins over Modified ZSM-5 Based Catalysts Stabilized by Hydrothermal Treatment" https://doi.org/10.3390/catal9050485).

Response #3

Thank you very much for your valuable comments. It is important to consider the aromatic cycle in order to understand the experimental results. We reviewed some papers regarding the MTH and DTO reaction mechanisms, and have added explanations corresponding to previous findings and experimental results in lines 4–10 in Page 3, and in the last two paragraphs in Page 11.

Comment #4

I think that the regression analysis needs to refer back to some of the work done by others to give it more context.

Response #4

Thank you for your comment. Three references regarding the regression analysis (Refs. 40– 42) were added to give a sentence on the importance of the regression analysis in the second paragraph in the Section 2.3. (Page 10).

Comment #5

Overall, this is a carefully carried out and written-up study of a series of commercially available zeolites, but it lacks mechanistic depth and explanation of the results. Given the solid experimental foundation of the study, I think that it could be accepted with major revisions to the discussion of the results and without carrying out any additional experiments. There are some comprehensive reviews, for example: ("Conversion of Methanol to Hydrocarbons: How Zeolite Cavity and Pore Size Controls Product Selectivity." https://doi.org/10.1002/anie.201103657.), which give excellent background on some of the discussions necessary to help explain the results.

Response #5

Thank you for your valuable comment. Refs. 44 and 46 regarding the reaction mechanism were added to the revised manuscript. The explanation regarding the expected reaction mechanism was added to the last two paragraphs in Page 11 and lines 1–3 in the second paragraph in Page 13, to address both the DTO and MTH reaction mechanisms.

Round 2

Reviewer 2 Report

Thank you for adding the mechanistic background, it could be more detailed but it is a big improvement to the discussion.

On line 45, what is the target yield for competitive economics?

Line 300 "olefin" is spelled incorrectly. Line 299 from “via the dealkylation”, the “the” is not necessary.

On line 110, I think to use the word “amount” to describe how acidic the zeolite is, is vague and not useful, I would rather a more technical and descriptive word.

“1,3-Butadiene (1,3-BD) is a raw material for general-purpose rubbers such as polybutadiene and styrene-butadiene rubbers. In 2018, approximately 15.3 billion kilograms of synthetic rubber was produced worldwide. Demand for 1,3-BD is expected to increase in the future because its addition to rubber and plastic can improve durability and flexibility.”

I think you could place the statement about the addition to rubber and plastic into the first statement after “…rubbers” and absorb the section about demand to make one nice sentence, this is quite an awkward first paragraph.

My comment regarding linking of sentences was not a criticism of the content, in fact I think that the sentences were important for context: it was simply a note about how they are linked and the flow of the text. It was also a general comment and those highlighted sentences were just an example, I apologise if that was not clear. Since I already commented on the brevity of the introduction, removal of those sentences about petrochemical-derived budadiene is making this problem worse, I would suggest that you put them back in with some editing to make it read well.

To this point, I do not think that my initial criticism of sentence length and linking has been addressed, the number of sentences which read as standalone statements are far too high. In many cases simply an “and” or something similar would suffice since many of these paragraphs read like bullet points, not prose. It is rather endemic I will not highlight each one, it often follows a pattern of a statement with the following sentences supporting the statement but with no linking words between them: from lines 291 to 315 is a great example of this where it occurs in each of the four paragraphs, although they are not the only examples. Lines 66 through 84 as well.

I would like to see some careful sentence conjugation to accept this to make it easier to read, and the lines mentioned which were taken out, and without any additional content I would be happy to accept it.

Author Response

Dear Reviewer 2,

We appreciate your comments about our revised manuscript and are delighted with your favorable response. We have revised the manuscript in accordance with your recommendations and hope that the changes that have been made are to your satisfaction.

Comments and Suggestions from Reviewer 2

Thank you for adding the mechanistic background, it could be more detailed but it is a big improvement to the discussion.

Comment #1

On line 45, what is the target yield for competitive economics?

Response #1

Thank you for your comment. The process simulation [6] indicated that the n-butene yield of 29.9 C-mol% led to the economics (balance of payment) competitive to the power generation. The target value was added to the line 55 in Page 2.

Comment #2

Line 300 "olefin" is spelled incorrectly. Line 299 from “via the dealkylation”, the “the” is not necessary.

Response #2

Thank you for your comment. “the olein cycle” was replaced with “the olefin cycle” in line 342 in Page 11. Line 341, “via the dealkylation” was replaced with “via dealkylation”.

Comment #3

On line 110, I think to use the word “amount” to describe how acidic the zeolite is, is vague and not useful, I would rather a more technical and descriptive word.

Response #3

Thank you for your comment. Lines 124-125 in Page 3, “the acid amount” was replaced with “the number of acid sites”.

Comment #4

“1,3-Butadiene (1,3-BD) is a raw material for general-purpose rubbers such as polybutadiene and styrene-butadiene rubbers. In 2018, approximately 15.3 billion kilograms of synthetic rubber was produced worldwide. Demand for 1,3-BD is expected to increase in the future because its addition to rubber and plastic can improve durability and flexibility.”

I think you could place the statement about the addition to rubber and plastic into the first statement after “…rubbers” and absorb the section about demand to make one nice sentence, this is quite an awkward first paragraph.

My comment regarding linking of sentences was not a criticism of the content, in fact I think that the sentences were important for context: it was simply a note about how they are linked and the flow of the text. It was also a general comment and those highlighted sentences were just an example, I apologize if that was not clear. Since I already commented on the brevity of the introduction, removal of those sentences about petrochemical-derived butadiene is making this problem worse, I would suggest that you put them back in with some editing to make it read well.

Response #4

Thank you for your valuable comment. We apologize for misunderstanding the reviewer's intentions. The statement about the addition of 1,3-BD to rubber and plastic was added to lines 30-33. The issue about the petrochemical-derived butadiene was added to the lines 34-37.

Comment #5

To this point, I do not think that my initial criticism of sentence length and linking has been addressed, the number of sentences which read as standalone statements are far too high. In many cases simply an “and” or something similar would suffice since many of these paragraphs read like bullet points, not prose. It is rather endemic I will not highlight each one, it often follows a pattern of a statement with the following sentences supporting the statement but with no linking words between them: from lines 291 to 315 is a great example of this where it occurs in each of the four paragraphs, although they are not the only examples. Lines 66 through 84 as well.

I would like to see some careful sentence conjugation to accept this to make it easier to read, and the lines mentioned which were taken out, and without any additional content I would be happy to accept it.

Response #5

Thank you for your valuable comment. We checked all the manuscript and reduced repetitive phrases/words.
